# Sulfonation Treatment of Polyether-Ether-Ketone for Dental Implant Uses

Hussein Hamid [1,*], Ihab Safi [1] and Falah Hussein [2]

1 Department of Prosthodontics, College of Dentistry, University of Baghdad, Baghdad 10071, Iraq; ihab.nsafi@codental.uobaghdad.edu.iq
2 Department of Pharmaceutics, College of Pharmacy, University of Babylon, Hilla 51002, Iraq; abohasan_hilla@yahoo.com
* Correspondence: hussein.kareem1101a@codental.uobaghdad.edu.iq

**Abstract:** There has been a recent uptake in the use of polyether-ether-ketone (PEEK), which is an organic thermoplastic polymer, in the manufacturing of various medical devices, implants, and equipment. Finding the best time and procedure for PEEK after sulfonation is the goal of this research. A total of 30 PEEK discs were sulfonated in this study by immersing them in concentrated ($H_2SO_4$) sulfuric acid for various durations and subsequently treated using various post-treatment techniques. Five experiments were carried out, aimed studying the effect of immersion time (5 s–2 min). The methods used as post-treatment were hydrothermal treatment, immersion in NaOH, and washing with acetone. The sulfonation time was measured, and the post-treatment techniques, surface characterizations, were conducted using scanning electron microscopy (SEM) (Electron Optics Instruments, LLC., West Orange, NJ, USA), atomic force microscopy (AFM) (AFM, Vía Burton, CA, USA), and hydrophilic properties. The results were confirmed by attenuated total reflectance-Fourier transform infrared spectroscopy (ATR-FTIR). The findings of this study demonstrate that sulfonating PEEK caused a structure with a porous network to form in every sample. As the sulfonation time increased, the porous structure became more noticeable and the concentration increased. As a consequence, the roughness of the surface increased notably, and the modified PEEK surface's wettability improved noticeably. Hydrothermal treatment was determined to be the most successful way for eliminating the leftover sulfuric acid, and sulfonation for 2 min was determined to be ideal. By understanding the best post-treatment procedures and ideal sulfonation duration, a theoretical foundation for the production of sulfonated PEEK for orthopedic uses may be laid.

**Keywords:** polyether-ether-ketone; surface modification; sulfonation; dental implant





## 1. Introduction

A new thermoplastic engineering plastic is polyether-ether-ketone (PEEK) [1]. Its mechanical, chemical, and biological qualities are outstanding [2,3]. Its main biomedical use is as an orthopedic implant in vivo. PEEK's biomechanical qualities are similar to those of human bones, reducing the danger of bone resorption and osteolysis induced by implant stress shearing. Medical PEEK is subject to fundamental performance criteria. The human body is tough and caustic [3,4].

Body fluids include several electrolyte ions and complex chemical substances. Chemical erosion is inevitable if metal implants are exposed in human fluids for a long period of time [5,6]. Therefore, orthopedic implants should resist rusting. The majority of corrosion is electrochemical. Electrochemical changes cause metal orthopedic implant pitting corrosion in humans. Function-based implant criteria are also important [7]. As sterile orthopedic implants, PEEK is stable, radiation-resistant, and mature. Ethylene oxide, gamma rays, and hot steam disinfect it [8].

PEEK's linear aromatic backbone contains functional ether and ketone groups [9]. PEEK is biocompatible, compatible with reinforcing materials, and has low solubility and

water absorption. Its corrosion resistance, fatigue resistance, high-temperature stability, sterilization stability, radiolucent, and ease of machinability are improved by these qualities.

Fixed crowns, fixed and detachable bridges, removable dentures, implant abutments, dental implants, and more are made from PEEK. Because of its better elastic and esthetic modulus, which are similar to those of human bones, PEEK has been regarded as a good contender to replace titanium in dental implants [10].

Ion-exchangeable charged groups are introduced to PEEK polymer chains with sulfuric acid to improve hydrophilicity. Sulfonation promotes polymer hydrophilicity and cation transport [11]. Acid etching modifies surface topography and creates a microinterlocking structure [12,13]. Etching using 98% sulfuric acid creates a porous surface where adhesives can penetrate, improving bond strength [14]. There is unanimity that alumina sandblasting and sulfuric acid etching improve PEEK adhesion [15].

Sulfonated PEEK may be prepared by understanding the optimal sulfonation duration and post-treatment procedure. This work examined the characteristics of sulfonated PEEK with five sulfonation periods (5, 10, 30, 60, and 120 s) and three post-treatment procedures (NaOH immersion, acetone washing, and $H_2O$ hydrothermal treatment). Finding the optimal sulfonation period and post-treatment procedure was the aim.

We hypothesized that pre-treatment with either mechanical and/or chemical means would result in the possible bonding of composite resin to PEEK. Secondly, as the time of immersion extended, the porous structure increasingly became more evident as well as more sophisticated.

## 2. Materials and Method

### 2.1. Preparing the Samples

Thirty disc-shaped samples of medical-grade PEEK were provided by Energetic Industry Co., Ltd., which is located in Beiliu, China. By cutting continuous extruded PEEK rods, disc-shaped samples with a 10 mm diameter and a 2 mm thickness [16] were created. For the in vitro surface characterization of sulfonated PEEK, these discs served as the test subject.

### 2.2. PEEK Sulfonation

The disc-shaped samples that were employed were put through the process. In order to determine the optimal times for sulfonation, five sets of different sulfonation times were tested: 5 s, 10 s, 30 s, 1 min, and 2 min. The untreated PEEK served as the control for this experiment. In order to produce sulfonated PEEK, discs made of PEEK were submerged in room-temperature solutions of concentrated $H_2SO_4$ for varying amounts of time (refer to Scheme 1 and Table 1 (including Images 1A–D)). Three different post-treatment procedures were conducted in order to maximize the process of removing any residual sulfuric acid: (1) rinsing in acetone for ten minutes while ultrasonic stirring was performed; (2) immersion in a solution of sodium hydroxide (NaOH) containing six weight percent of sodium hydroxide (NaOH) for five minutes; and (3) hydrothermal treatment at a temperature of one hundred and twenty degrees Celsius for four hours.

### 2.3. Surface Characterization

The surfaces of untreated PEEK, which served as the control, and sulfonated PEEK were analyzed using the following criteria:

1. All of the analyses of the sulphonated PEEK's' chemical composition were carried out at room temperature and using attenuated total reflection-Fourier transform infrared (ATR-FTIR). (IRAffinity-1, ATR-FTIR Shimadzu, Shimadzu Scientific Instruments, Kyoto, Japan) was used to obtain the transmittance spectrum recorded from 4000 to 600 cm$^1$ in order to validate the presence of new sulfonated groups with PEEK polymer chains after the treatments.

2. Scanning electron microscopy (SEM) (SEM Test Speed Vega 111, Electron Optics Instruments, LLC., West Orange, NJ, USA) was utilized in order to evaluate the surface

morphology. For the purpose of elemental analysis, a scanning electron microscope with energy dispersive X-ray (EDX) analysis was carried out.

3. Conduction of a test of wettability. The wettability of the samples was measured using a contact angle goniometer made by Creating Nano Technologies Inc. in Taiwan, China (model number Cam110). In order to assess the surface hydrophilicity of sulphonated PEEK at room temperature, the static contact angle measurement was utilized to quantify the water contact angle of the material. The water contact angle on a PEEK surface that had not been treated in any way served as the control for this experiment.

4. The atomic force microscopy, also known as the AFM. In the tapping mode, a contact AFM (BenYuan CSPM-5500, Being Nano-Instruments Ltd., Beijing, China) was utilized to produce a 3D topographical picture, as well as the surface area ratio (Sdr) for the selected sulphonated PEEK and the average surface roughness (Sa) in nanometers. Additionally, the average surface roughness was measured in nanometers [17].

**Scheme 1.** Sulfonation of the PEEK and neutralization of the sulfonated PEEK by rinsing in acetone, immersion in a solution of sodium hydroxide (NaOH), and hydrothermal treatment at 120 C.

**Table 1.** Sulfonation times/s.

| Samples | Immersion Solution | Sulfonation Time/s | Observation | SEM Test |
|---|---|---|---|---|
| A | | | | As the control, with a flat and smooth surface (Image 1A) |
| B, C, D, E | Concentrated $H_2SO_4$ | 5, 10, 30, 60, and 120 s | | The porous structure gradually became more obvious (Image 1B–E) |
| F | Concentrated $H_2SO_4$ | More than 120 s | | Melting and damage of the whole sample. These sample were neglected |

*2.4. Statistical Analysis*

For data analysis, Prism 9 (GraphPad Software version 9.0, USA) and SPSS (Statistical Package for Social Science, version 21) were utilized. For the purpose of descriptive analysis, the findings are shown as bar charts with mean values and standard deviations. A one-way ANOVA and the post hoc Tukey's HSD test were used. *p*-values of more than 0.05, less than 0.05, and less than 0.01 indicated non-significant, significant, and highly significant differences, respectively.

**3. Results**

*3.1. Washing and Removal Methods of the Acid Residues*

The FTIR spectra of various sulfonated PEEKs with varying degrees of sulfonation over time are displayed in Figure 1. According to [18], the carbon–carbon stretching vibrations in the aromatic ring caused peaks in the frequency ranges of 1600–1585 $cm^{-1}$ and 1500–1400 $cm^{-1}$ in all of the samples. In comparison to the spectra of the untreated PEEK, the novel sulfonate substitution resulted in the appearance of additional peaks between the wavelengths of 1138.00 and 1473.62 $cm^{-1}$. The findings demonstrate that functional groups of $SO_3H$ were incorporated into the polymer chains of PEEK through the process of sulfonation.

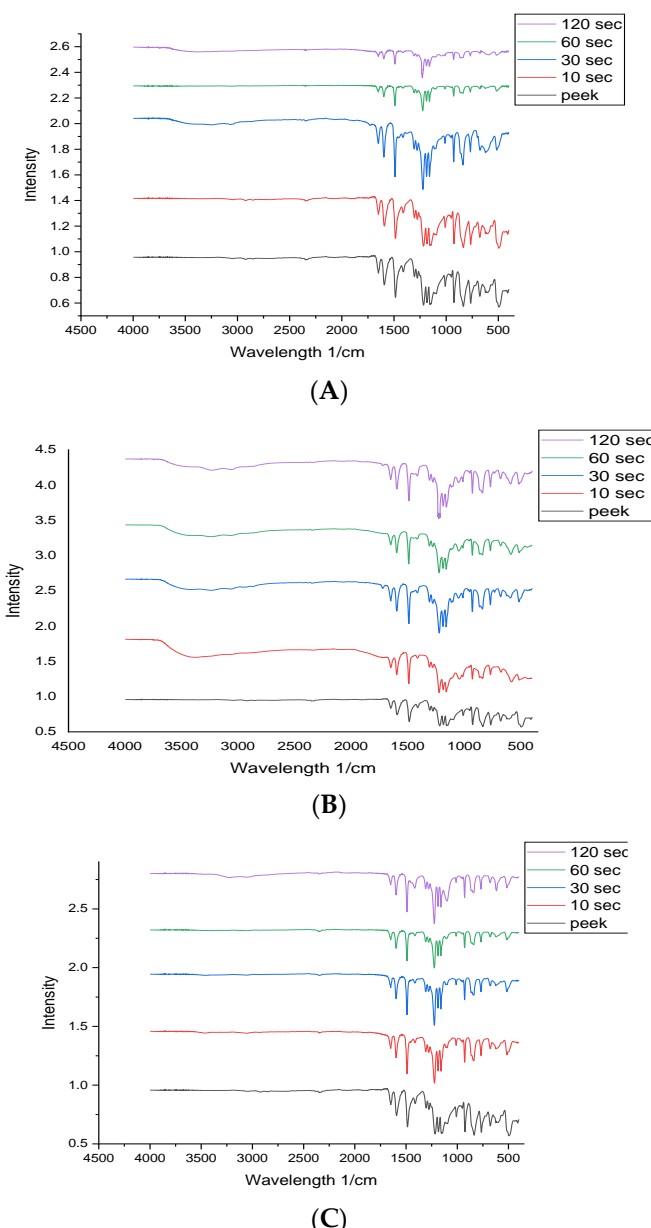

**Figure 1.** The FTIR of the sulfonated PEEKs with various times of sulfonation (5, 10, 30, 60, and 120 s) and three post-treatment procedures: (**A**) NaOH immersion, (**B**) acetone washing, and (**C**) $H_2O$ hydrothermal treatment.

*3.2. Chemical Characterization and Surface Morphology of the Sulfonated PEEKs with Various Sulfonation Times*

The pictures obtained with SEM and AFM are shown in Figures 2 and 3, respectively. Before it was sulfonated, the PEEK's surface was perfectly flat and smooth (Figure 2A). The sulfonation of PEEK with strong sulfuric acid resulted in the formation of a three-dimensional porous network in all of the sulfonated PEEK samples (Figure 2B–D).

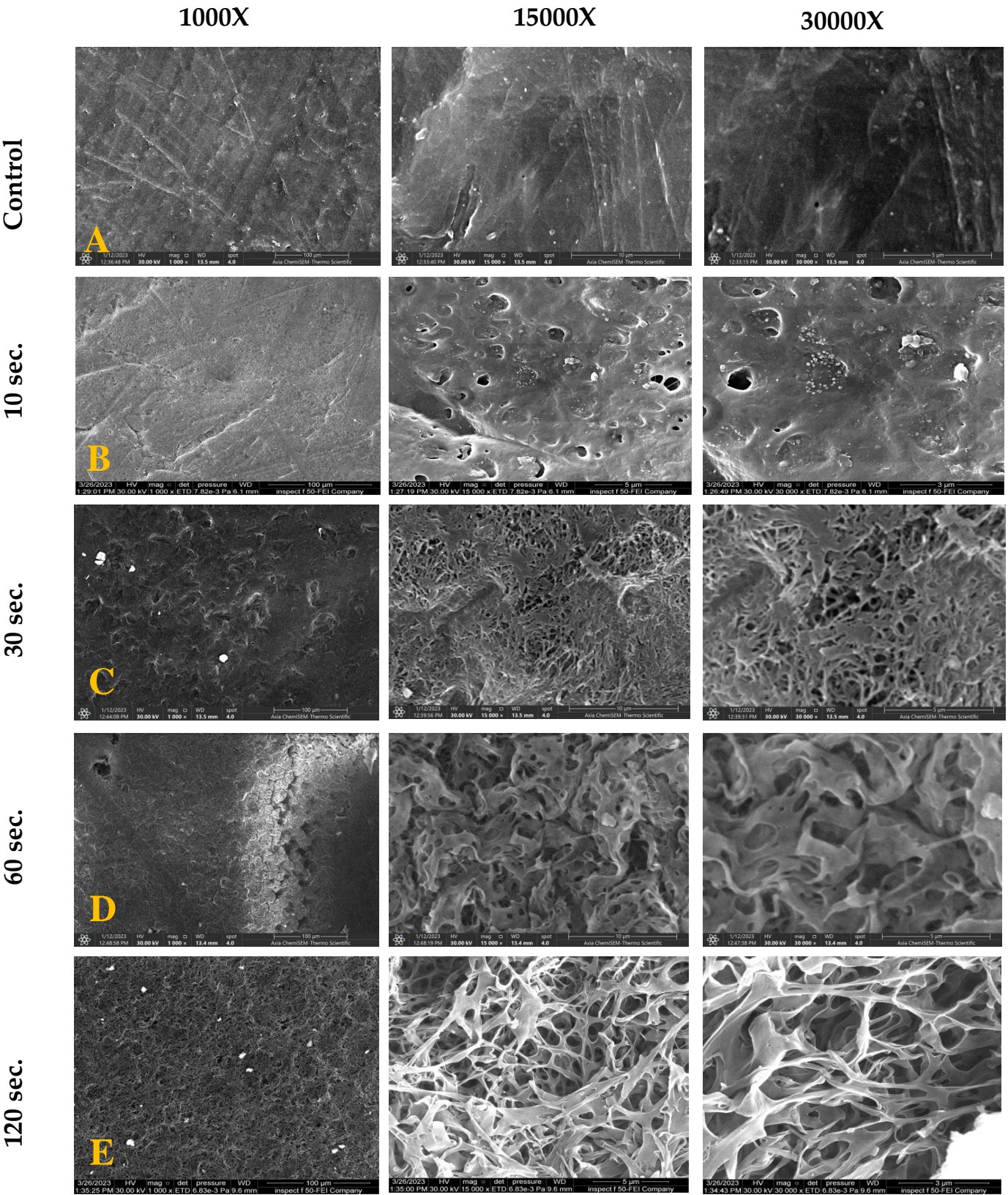

**Figure 2.** SEM characterization of the surface morphology of PEEK and sulfonated PEEK samples with different sulfonation times. At different magnifications: 1000×, 15,000×, and 30,000×.

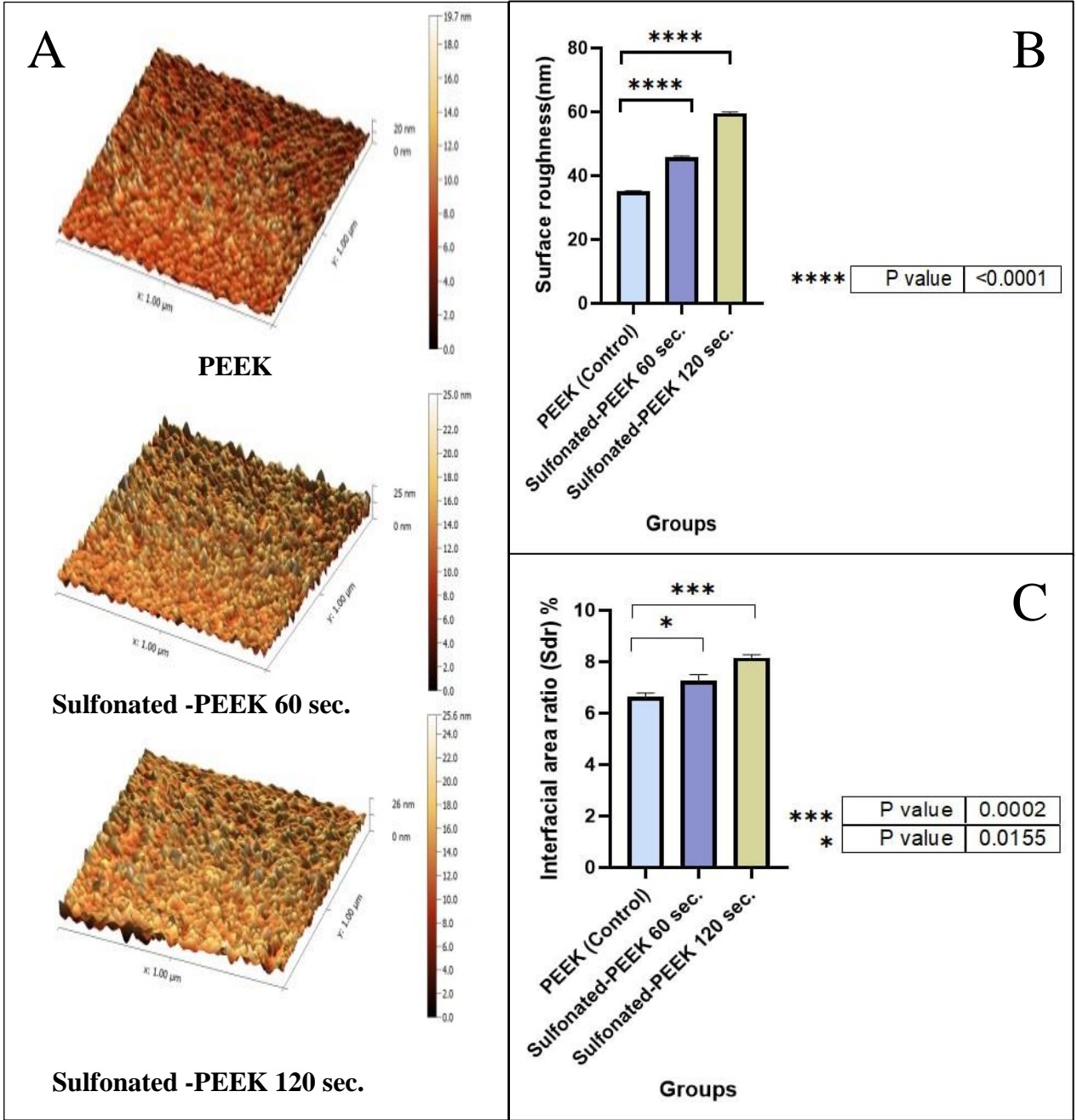

**Figure 3.** Surface morphology characterized by AFM for PEEK and sulfonated PEEK samples with various sulfonation times. (**A**) Three-dimensional topographical images; (**B**) mean value of the surface roughness (Sa), with Turkey's HSD significance; and (**C**) mean value of the interfacial area ratio (Sdr), with Turkey's HSD significance.

The location and quantity of sulfur contained in the specimens were further investigated through the use of EDX mapping and analysis. EDX mapping at multiple locations validated the sulfur distribution (Figure 4).

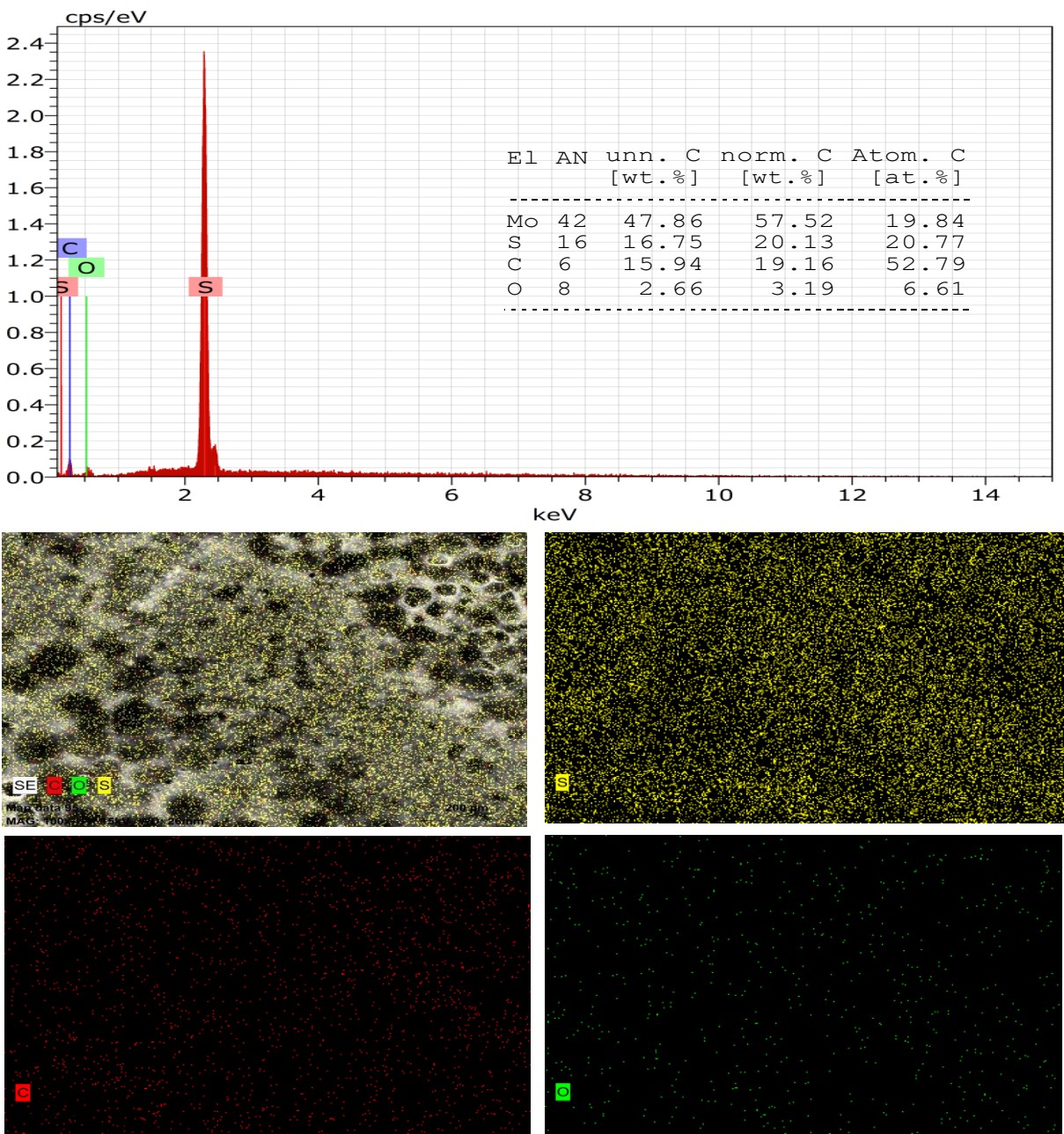

**Figure 4.** SE: EDX mapping of sulfonated PEEKs with 120 s sulfonation and $H_2O$ hydrothermal post-treatment procedures, showing the distribution of the SE S, O, and C elements.

### 3.3. Sulfonated PEEKs' Hydrophilic Properties with Various Sulfonation Times

An important factor to consider when conducting research on the hydrophilic qualities of a material's surface is the contact angle. Table 2 and Figure 5 present the findings of the investigation on the water contact angles. In each group, the average water contact angles of sulfonated PEEKs were lower than that of PEEK (90.00). For example, the water average contact angles of 5 s sulfonated PEEK were 87.227 and 90.00, and the average water contact angles of 30 s sulfonated PEEK were 81.467 and 75.212. The untreated PEEK exhibited a contact angle that was greater than that of the other groups, which suggests that sulfonation enhances PEEK's hydrophilicity.

**Table 2.** The etching times and angles obtained.

| Sample | Etching Times | Angles Obtained |
|---|---|---|
| PEEK (control) | | 90 |
| Sulphonated PEEK (5 s) | 5 s | 90.00 |
| Sulphonated PEEK (10 s) | 10 s | 90.00 |
| Sulphonated PEEK (30 s) | 30 s | 87.22 |
| Sulphonated PEEK (60 s) | 60 s | 81.467 |
| Sulphonated PEEK (120 s) | 120 s | 75.21 |

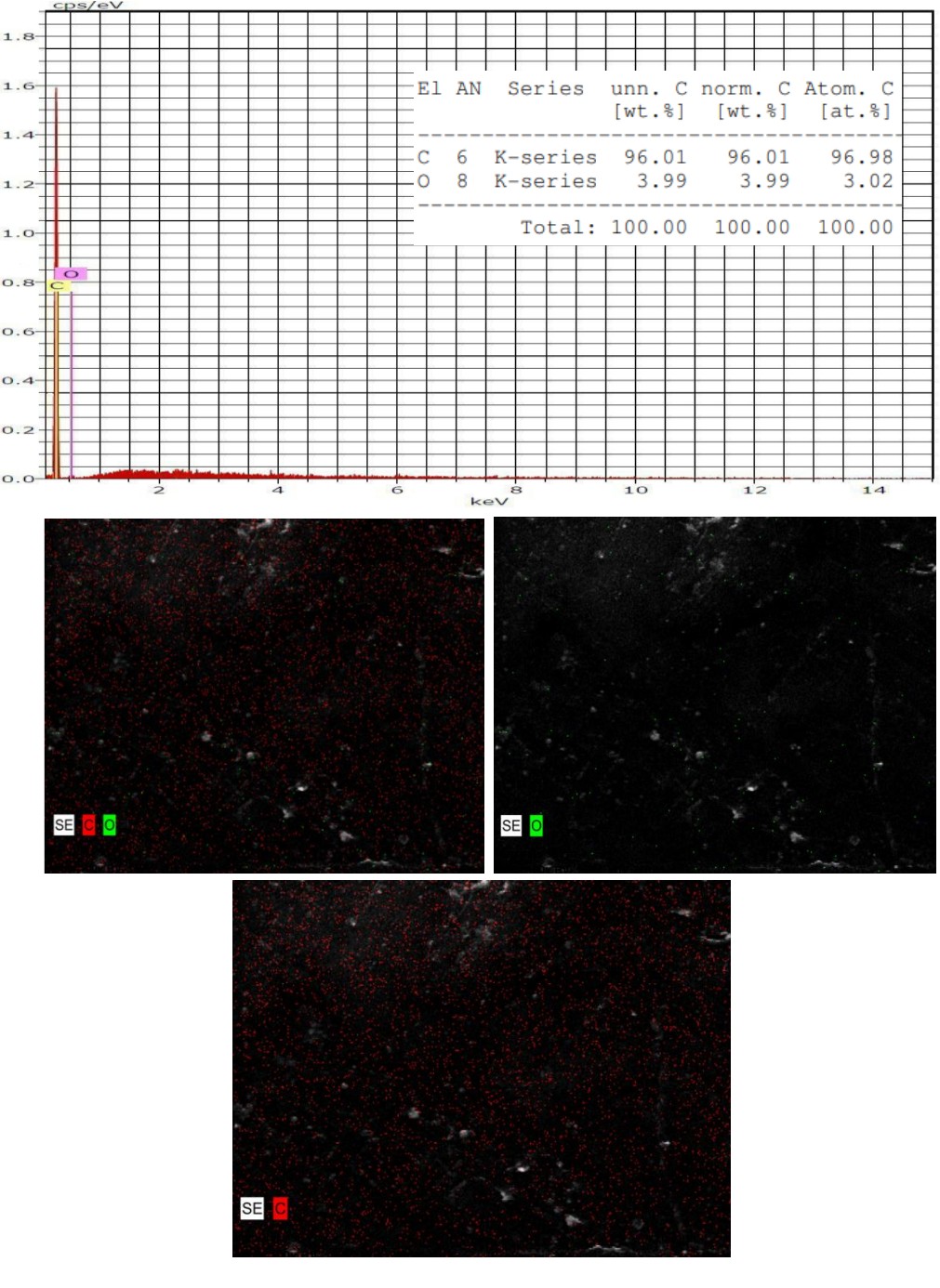

(A)

**Figure 5.** *Cont.*

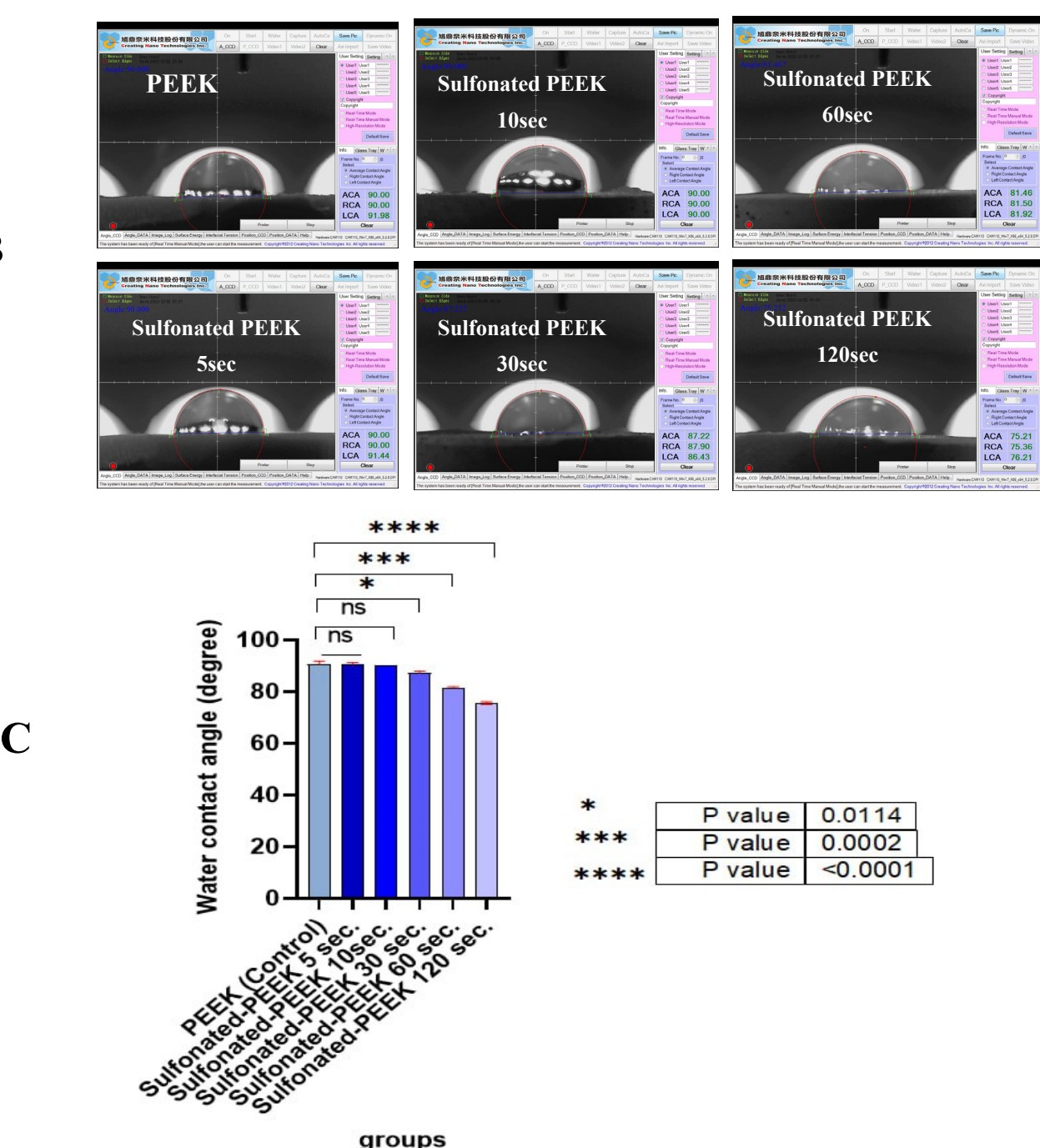

**Figure 5.** (**A**): EDX mapping of PEEK, showing the distribution of the O and C elements only. (**B**) Water contact angle measurements. (**C**) Water contact angle's mean values, with Turkey's HSD significance.

As the result of the current study, the significance of chemical processes on the polymer was emphasized. Furthermore, the complexity of interaction and surface topography were also considered.

The operation steps of the sulfonated surface modification method are relatively simple and can form a porous structure conducive to cell growth on the PEEK surface, introduce sulfonic acid groups with biological activity, and improve PEEK's antibacterial and osteogenic capabilities. However, the concentrated sulfuric acid and other modified reagents are dangerous to handle; so, the experiment has certain limitations.

PEEK can overcome some of the limitations of metal implants, such as stress shielding and metal allergy. PEEK is currently used as an orthopedic implant in clinical practice and has achieved good results.

## 4. Discussion

Acetone rinsing was the cleaning method used in this study in order to remove the sulfuric acid's residues in the porous network structure without affecting the surface shape or the chemical composition of the sulfonated PEEK since acetone does not dissolve sulfonated PEEK [19]. Since PEEK is a crystalline polymer with a melt transition temperature of approximately 343 degrees Celsius and a semicrystalline thermoplastic polymer with a glass transition temperature of about 143 degrees Celsius [18], the hydrothermal treatment at a temperature of 120 degrees Celsius can only remove the sulfuric acid that is left behind. This indicates that neither its physical structure nor its chemical composition can be altered by the hydrothermal treatment that is being applied. The residual sulfuric acid on the sulfonated PEEK seeped from the pores into the alkaline solution, and after immersion in the NaOH solution, the solutions were neutralized by each other; the pores' structure was not impacted in the slightest. After being sulfonated, PEEK should be treated with one of the following three procedures: washing with acetone, immersing in NaOH, or heating with heat. Any of these three processes is recommended as an effective PEEK process treatment since it may eliminate any residual sulfuric acid.

As the time of immersion extended, the porous structure increasingly became more evident as well as more sophisticated. At first, the porous structure on the surface of the sulfonated PEEK was just superficial and straightforward, but as the time increased, it gradually became more complex. After undergoing sulfonation in concentrated $H_2SO_4$, the surface of the sulfonated PEEK exhibited a three-dimensional nano- to micro-porous network. Additionally, sulfonated functional groups were discovered on the surface of the sulfonated PEEK. When the sample was removed from the sulfuric acid, there was a trace amount of $H_2SO_4$ that was left on the surface of the porous material.

On the other hand, if the duration of the immersion was prolonged for an excessive amount of time, the porous structure that was produced on the surface layer had a tendency to be damaged and destroyed. According to the findings of this research project, as the amount of time spent sulfonating increased, the structure of the porous network became more apparent. In a previous work, [20] found that increasing the immersion duration resulted in the improved formation of nanostructures and porous materials. The findings of this investigation are in agreement with the findings of previous investigations.

The findings demonstrated that the sulfur element was incorporated into the polymer chains of PEEK through the process of sulfonation.

In terms of hydrophilicity, the untreated PEEK's water contact angle was found to range from 70 to 90 degrees in previous research [20], which is comparable to our findings. According to the findings of earlier research conducted by [21], the PEEK's water contact angle was measured as 78.6, whereas the sulfonated PEEK's water contact angle was measured as 67.2. Although hydrophilic $SO_3H$ groups were added to the surface, the findings of this study suggest that the morphology of the surface with a nanostructured network and 3D porosity plays a crucial role in diminishing the hydrophilicity of the surface. This was the conclusion drawn from this study. According to [5,20], the bioactivity and osseointegration of a bone implant interface are significantly improved when the bone implant interface possesses hydrophilicity and nano-topography. The sulfonation of PEEK results in a considerable reduction in the material's surface hydrophilicity; yet, the 3D nano- to micro-porous structure that is created as a result of sulfonation is thought to be favorable for increasing the bioactivity.

However, contrary findings were found in other research [22]. These findings concerned an increase in the contact angle following sulfonation. The authors of [23] showed that the surface morphology of sulphonated PEEK as well as the hydrophilic sulfonate functional groups play a major part in increasing the hydrophilicity of sulfonated PEEK.

Despite the fact that many studies have reported various results, this research has shown that both of these factors have a vital impact.

Within the scope of this research, a rise in surface roughness was documented as a probable concurrent impact. As a result, the adhesive qualities of the surface were shown to increase even more. It is feasible to develop oxygen-rich nanofilms on PEEK with a high surface energy, which can result in an increased cell performance [24,25].

Along with the surface's chemistry, the surface's topography also has a significant impact on the behavior of cells when they are attached to the surface [26,27]. The mechanical interlocking that occurs as a result of nanoscale surface roughness can lead to improved bone implant fixation. The surface roughness at the nanoscale, on the other hand, has been shown to dramatically alter hydrophilicity, which in turn has the potential to drastically affect the behavior of cells. When cells are cultivated on substrates of varying roughness, they frequently take on distinct forms. There is a wealth of data to suggest that the shapes of the cells are connected to their behaviors, such as proliferation and protein secretion [27–29].

## 5. Conclusions

The ideal sulfonation duration was determined to be 2 min since a porous structure was created very effectively and there was a comparatively low amount of leftover sulfuric acid. The hydrothermal treatment had a greater impact on eliminating the remaining sulfuric acid and had no impact on modifying the morphology of the surface when compared to NaOH immersion or acetone washing. This is due to the fact that the hydrothermal treatment uses higher temperatures. As a result, the hydrothermal treatment of water is recommended for the purpose of treating sulfonated PEEK. A theoretical foundation may be created for the preparation of sulfonated PEEK in order to achieve a potentially advantageous impact by defining the ideal sulfonation period and post-treatment procedure.

Polyether-ether-ketone and its composites play an important role in the field of oral repair given their excellent physical and chemical properties as well as biological properties. Its machinability enables the accurate manufacturing of various kinds of implants with complex structures. Its excellent properties, such as stable chemical properties, good biosecurity, and elastic modulus, are close to those of human dense bones, and therefore, make it an excellent potential oral implant material. The biological modification methods described above are all aimed at improving PEEK's biological activity. Each modification method has its advantages and disadvantages. Despite considerable experimental studies, modification technologies are still immature and lack sufficient clinical data to prove the clinical efficacy of these modification methods. Therefore, future research should focus on the development of more efficient and practical modification methods and clinical practice, and the exploration of PEEK modification methods to address different branches of stomatology.

**Author Contributions:** Conceptualization, H.H.; Methodology, H.H. and I.S.; Resources, H.H.; Writing—original draft, H.H.; Writing—review & editing, I.S.; Supervision, I.S. and F.H. All authors have read and agreed to the published version of the manuscript.

**Funding:** This research received no external funding.

**Institutional Review Board Statement:** Not applicable.

**Informed Consent Statement:** Not applicable.

**Data Availability Statement:** The original contributions presented in the study are included in the article, further inquiries can be directed to the corresponding author.

**Conflicts of Interest:** The authors declare no conflicts of interest.

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
