# Peer review of "Sulfonation Treatment of Polyether-Ether-Ketone for Dental Implant Uses"

_applsci, doi:10.3390/app14103980_

Round 1
Reviewer 1 Report
Comments and Suggestions for Authors
Dear Authors,
Congratulations on the job you have done and presented in this manuscript.
Unfortunately, in the present form the manuscript cannot be published in a such a high prestigious journal. There are multiple issues that need to be revised prior to consideration for publication.
First and foremost, there are multiple spelling and grammar issues, English needs to be revised. There are multiple text editing and format errors, figures overlapping, not cited in text. The methodology was not conducted properly....Please see the attachment

Moderate changes required
Author Response
Thank you for giving us the opportunity to submit a revised draft of the manuscript “Sulfonation Treatment of Polyether-ether-ketone for Dental Implant Uses” for publication in the Applied Sciences. We appreciate the time and effort that you and the reviewers dedicated to providing feedback on our manuscript and are grateful for the insightful comments on and valuable improvements to our paper. We have incorporated most of the suggestions made by the reviewers. Those changes are highlighted within the manuscript. Please see below, in red, for a point-by-point response to the reviewers’ comments and concerns.

Reviewer 2 Report
Comments and Suggestions for Authors
Sulfonation Treatment of Polyether-ether-ketone for Dental 2 Implant Uses
In this study, the optimal timing of post-sulfonation procedures on PEEK material is investigated. Longer sulfonation resulted in increased porous structure and improved surface wettability. Hydrothermal treatment was found to be the most effective for sulfuric acid removal. Sulfonation for 2 minutes was identified as ideal.
Concerns:
Page 1, lines 34 and 35: There is a typographical error, the authors have confused the word "shield" with the word "shear". “reducing the danger of bone resorption and osteolysis induced by implant stress shielding.” Instead “reducing the danger of bone resorption and osteolysis induced by implant stress shearing.”
Page 2: there is a contradiction with the test groups. The authors first refer on page 2 line 61 to 5 groups "(5, 10, 30, 60, and 120 sec)" but later on page 2 lines 75 and 76 they mention 8 groups "5 seconds, 10 seconds, 30 seconds, 1 minute, 2 minutes, 3 minutes, 4 minutes, and 5 minutes". How many groups were created?
Page 5, line 174: The authors refer in the following heading to the different sulfonated groups "3.1 Sulfonated PEEKs' chemical characterization with various sulfonation times" However, in the development they explain the methods of washing and removal of the acid residues. This heading should be in accordance with the text.
Page 12, lines 247-249: it is not clear to which angles each sulfonation time corresponds. Since a surface is considered hydrophobic when its contact angle is between 90-120º and hydrophilic when it is between 10-90º, it does not seem that there is a clear improvement. I suggest including a table with the etching times and angles obtained with their confidence intervals.
Page 12, line 265: all abbreviations must be defined before their first use "SPEEK" has not been defined (we assume it is sulfonated PEEK).
Page 13 line 290: This is not a conclusion of your study and does not follow from your work "PEEK can be sulfonated to boost its bioactivity" It should be deleted.
Author Response

(The authors gave the same response as above.)

Round 2
Reviewer 1 Report
Comments and Suggestions for Authors
Dear authors ,
I believe that you took advantage on the reviewer's comments and improved the manuscript. I have no further comments.